# DExT: Detector Explanation Toolkit

## Abstract

State-of-the-art object detectors are treated as black boxes due to their highly non-linear internal computations. Even with unprecedented advancements in detector performance, the inability to explain how their outputs are generated limits their use in safety-critical applications. Previous work fails to produce explanations for both bounding box and classification decisions, and generally make individual explanations for various detectors. In this paper, we propose an open-source Detector Explanation Toolkit (DExT) which implements the proposed approach to generate a holistic explanation for all detector decisions using certain gradient-based explanation methods. We suggests various multi-object visualization methods to merge the explanations of multiple objects detected in an image as well as the corresponding detections in a single image. The quantitative evaluation show that the Single Shot MultiBox Detector (SSD) is more faithfully explained compared to other detectors regardless of the explanation methods. Both quantitative and human-centric evaluations identify that SmoothGrad with Guided Backpropagation (GBP) provides more trustworthy explanations among selected methods across all detectors. We expect that DExT will motivate practitioners to evaluate object detectors from the interpretability perspective by explaining both bounding box and classification decisions.

## 1 Introduction

Object detection is imperative in applications such as autonomous driving (Feng et al., 2021), medical imaging (Araújo et al., 2018), and text detection (He et al., 2017). An object detector outputs bounding boxes to localize objects and categories for objects of interest in an input image. State-of-the-art detectors are deep convolutional neural networks (Zou et al., 2019), a type of Deep Neural Network (DNN), with high accuracy and fast processing compared to traditional detectors. However, convolutional detectors are considered black boxes (Shwartz-Ziv & Tishby, 2017) due to over-parameterization and hierarchically non-linear internal computations. This non-intuitive decision-making process restricts the capability to debug and improve detection systems. The user trust in model predictions has decreased and consequently using detectors in safety-critical applications is limited. In addition, the process of verifying the model and developing secure systems is challenging (Doshi-Velez & Kim, 2017; Zablocki et al., 2021). Numerous previous studies state interpreting detectors by explaining the model decision is crucial to earning the user's trust (Wagstaff, 2012; Rudin & Wagstaff, 2014; Spiegelhalter, 2020), estimating model accountability (Kim & Doshi-Velez, 2021), and developing secure object detector systems (Doshi-Velez & Kim, 2017; Zablocki et al., 2021).

With a range of users utilizing detectors for safety critical applications, providing humanly understandable explanations for the category and each bounding box coordinate predictions together is essential. In addition, as object detectors are prone to failures due to non-local effects (Rosenfeld et al., 2018), the visualization techniques for detector explanations should integrate explanations for multiple objects in a single image at the same time. Previous saliency map-based methods explaining detectors (Petsiuk et al., 2021; Tsunakawa et al., 2019; Gudovskiy et al., 2018) focus on classification or localization decisions individually, not both at the same time.

In this paper we consider three deficits in the literatureL methods to explain each category and bounding box coordinate decision made by an object detector, visualizing explanations of multiple bounding boxes into the same output explanation image, and a software toolkit integrating the previously mentioned aspects.

This work concentrates on providing individual humanly understandable explanations for the bounding box and classification decisions made by an object detector for any particular detection, using gradient-based saliency maps. Figure 1 provides an illustration of the proposed solution by considering the complete output information to generate explanations for the detector decision.

Explanations for all the decisions can be summarized by merging the saliency maps to achieve a high-level analysis and increasing flexibility to analyze detector decisions, improving improving model transparency and trustworthiness. We suggest methods to combine and visualize explanations of different bounding boxes in a single output explanation image as well as an approach to analyze the detector errors using explanations.

This work contributes:

- DExT, software toolkit, to explain each decisions (bounding box regression and object classification jointly), evaluate explanations, and identify errors made by an object detector.

- A simple approach to extend gradient-based explanation methods to explain bounding box and classification decisions of an object detector.

- An approach to identify reasons for the detector failure using explanation methods.

- Multi-object visualization methods to summarize explanations for all output detections in a single output explanation.

- An evaluation of gradient-based saliency maps for object detector explanations, including quantitative results and a human user study.

We believe our work reveals some major conclusions about object detector explainability. Overall quantitative metrics do not indicate that a particular object detector is more interpretable, but visual inspection of explanations indicates that recent detectors like EfficientDet seem to be better explained using gradient-based methods than older detectors (like SSD or Faster R-CNN, shown in Figure 2), based on lack of artifacts on their heatmaps. Detector backbone has a large impact on explanation quality (Shown in Figure 6).

The user study (Section 4.4) reveals that humans clearly prefer the convex polygon representation, and Smooth Guided Backpropagation provides the best object detector explanations, which is consistent with quantitative metrics. We believe these results are important for practitioners and researchers of object detection interpretability, and the overall message is to explain both object classification and bounding box decisions, and it is possible to combine all explanations into a single image using the convex polygon representation of the heatmap pixels.

## 2 Related Work

Interpretability is relatively underexplored in detectors compared to classifiers. There are post hoc (Petsiuk et al., 2021; Tsunakawa et al., 2019; Gudovskiy et al., 2018) as well as intrinsic (Kim et al., 2020; Wu & Song, 2019) detector interpretability approaches. Detector Randomized Input Sampling for Explanation (D-RISE) (Petsiuk et al., 2021) in a model-agnostic manner generates explanations for detector decisions for the complete detector output. However, saliency map quality depends on the computation budget, the method is time consuming, and individual explanations for bounding boxes are not evaluated. Contrastive Relevance Propagation (CRP) (Tsunakawa et al., 2019) extends Layer-wise Relevance Propagation (LRP) (Bach et al., 2015) to explain individually the bounding box and classification decisions of Single Shot MultiBox Detector (SSD). This procedure includes propagation rules specific to SSD. Explain to fix (E2X) (Gudovskiy et al., 2018) contributes a framework to explain the SSD detections by approximating SHAP

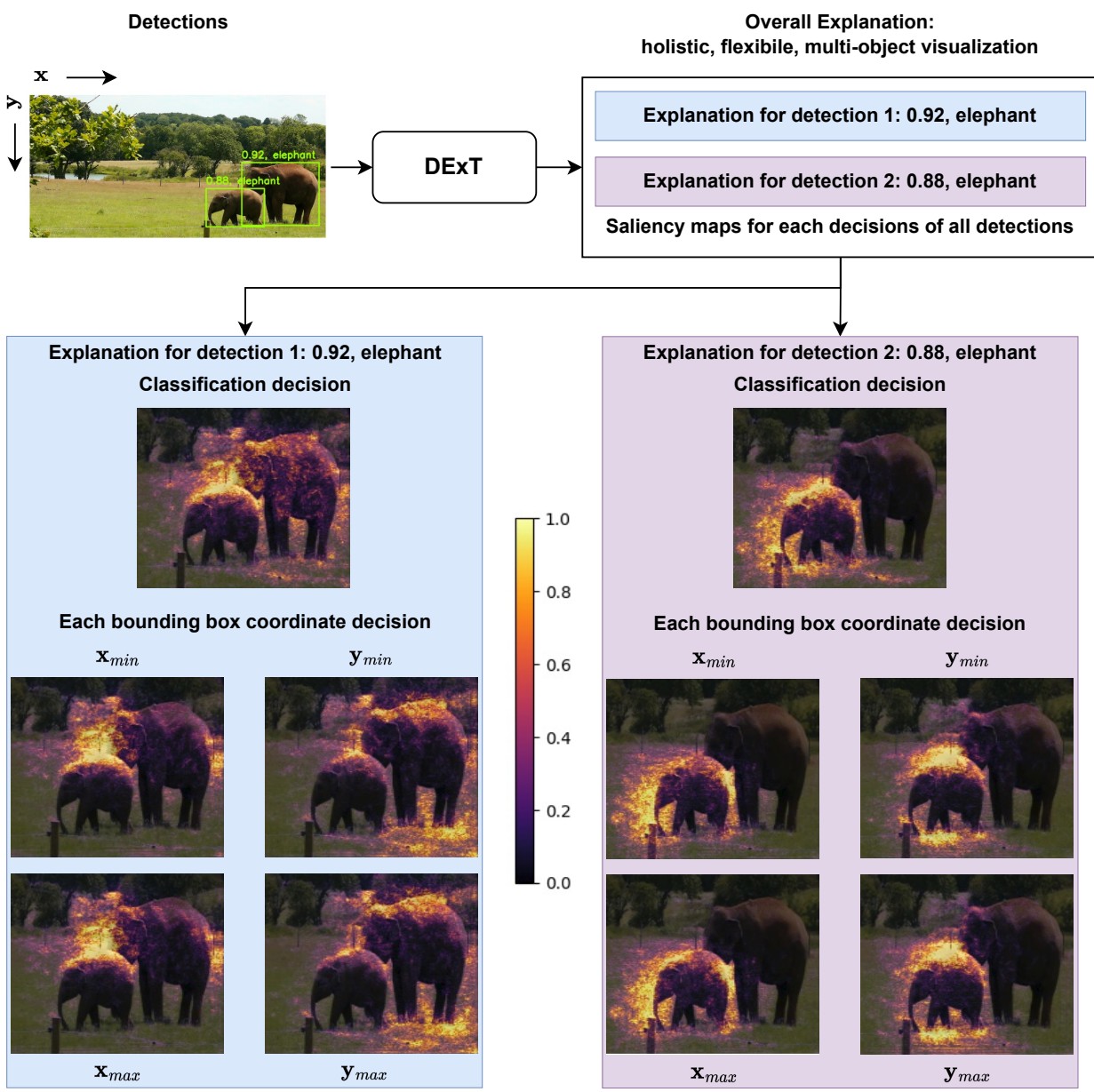

Figure 1: A depiction of the proposed approach to interpret all object detector decisions. The two elephants detected by the EfficientDet-D0 detector in the image have 0.92 and 0.88 confidence. The input image is taken from the MS COCO dataset (Lin et al., 2014). The corresponding explanations are provided in the same colored boxes. This breakdown of explanations offers more flexibility to analyze decisions and serves as a holistic explanation for all the detections. The explanation for each decision is a saliency map highlighting the important pixels of the input image. Saliency maps are overlaid on the input image to illustrate the correspondence with input image pixels.

(Lundberg & Lee, 2017) feature importance values using Integrated Gradients (IG), Local Interpretable Model-agnostic Explanations (LIME), and Probability Difference Analysis (PDA) explanation methods. E2X identifies the detection failure such as false negative errors using the explanations generated. The individual explanations for bounding box decisions and classification decisions are unavailable.

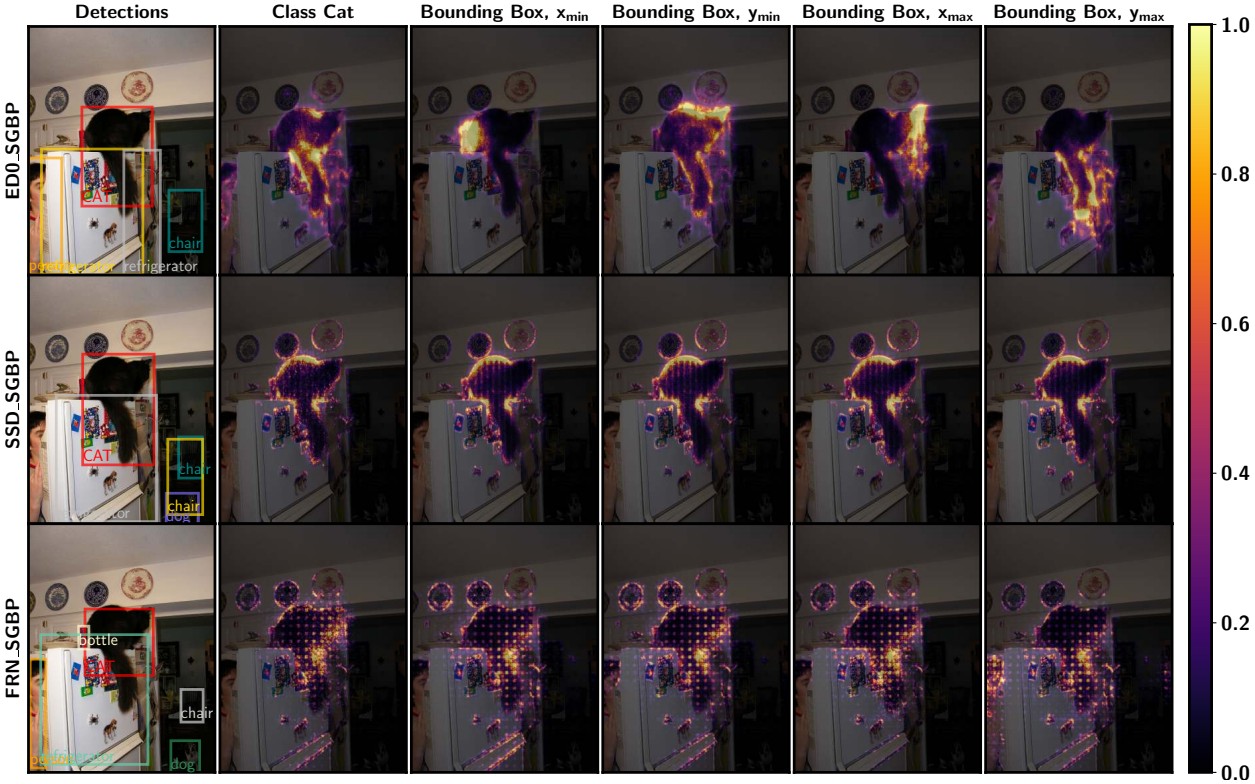

Figure 2: Comparison of the classification and all bounding box coordinate explanations corresponding to the cat detection (red-colored box) across different detectors using SGBP is provided. The bounding box explanations from EfficientDet-D0 illustrate the visual correspondence to the respective bounding box coordinates. The explanations from Faster R-CNN illustrate a sharp checkerboard pattern.

The intrinsic approaches majorly focus on developing detectors that are inherently interpretable. Even though the explanations are provided for free, currently, most of the methods are model-specific, do not provide any evaluations on the explanations generated, and includes complex additional designs.

Certain attention-based models such as DEtector TRansformer (DETR) (Carion et al., 2020) and detectors using non-local neural networks (Wang et al., 2018) offer attention maps improving model transparency. A few previous works with attention reveal contradicting notions of using attention for interpreting model decisions. Serrano & Smith (2019) and Jain & Wallace (2019) illustrate attention maps are not a reliable indicator of important input region as well as attention maps are not explanations, respectively. Bastings & Filippova (2020) have revealed saliency methods provide better explanations over attention modules.

In this work, post hoc gradient-based explanation methods are selected because the methods provide better model translucency, computational efficiency, do not affect model performance, and utilize the gradients in DNNs. Finally, saliency methods are widely studied in explaining DNN-based models (Ancona et al., 2019). A detailed comparative evaluation of various detectors reporting robustness, accuracy, speed, inference time as well as energy consumption across multiple domains has been carried out by Arani et al. (2022). In this work, the authors compare detectors from the perspective of explainability.

## 3 Proposed Approach

### 3.1 Explaining Object Detectors

This work explains various detectors using gradient-based explanation methods as well as evaluate different explanations for bounding box and classification decisions. The selected detectors are: SSD512 (SSD) (Liu et al., 2016), Faster R-CNN (FRN) (Ren et al., 2017), and EfficientDet-D0 (ED0) (Tan et al., 2020). The short-form tags are provided in the bracket. SSD512 and Faster R-CNN are widely used single-stage and two-stage approaches, respectively. Explaining the traditional detectors will aid in extending the explanation procedure to numerous similar types of recent detectors. EfficientDet is a relatively recent state-of-the-art single-stage detector with higher accuracy and efficiency. It incorporates a multi-scale feature fusion layer called a Bi-directional Feature Pyramid Network (BiFPN). EfficientDet-D0 is selected to match the input size of SSD512. The variety of detectors selected aids in evaluating the explanation methods across different feature extractors such as VGG16 (SSD512), ResNet101 (Faster R-CNN), and EfficientNet (EfficientDet-D0). The gradient-based explanation methods selected in this work to explain detectors are: Guided Backpropagation (GBP) (Springenberg et al., 2015), Integrated Gradients (IG) (Sundararajan et al., 2017), SmoothGrad (Smilkov et al., 2017) + GBP (SGBP), and SmoothGrad + IG (SIG). The short-form tags are provided in the bracket. GBP produces relatively less noisy saliency maps by obstructing the backward negative gradient flow through a ReLU. In addition, GBP is a simple and widely-used approach compared to other methods. For instance, an uncertainty estimate of the most important pixels influencing the model decisions is carried out using GBP and certain uncertainty estimation methods (Wickstrøm et al., 2020). This combines uncertainty estimation and interpretability to better understand DNN model decisions. IG satisfies the implementation and sensitivity invariance axioms that are failed by various other state-of-the-art interpretation methods. SmoothGrad aids in sharpening the saliency map generated by any interpretation method and improves the explanation quality. These four explanation methods explain a particular detector decision by computing the gradient of the predicted value at the output target neuron with respect to the input image.

The object detector decisions for a particular detection are bounding box coordinates ($x_{\min}$, $y_{\min}$, $x_{\max}$, $y_{\max}$), and class probabilities ($c_1, c_2, ..., c_k$), where $k$ is the total number of classes predicted by the detector. Usually these are output by heads at the last layer of the object detector. The classification head is denoted as $\text{model}_{\text{cls}}(x)$, while the bounding box regression head is $\text{model}_{\text{bbox}}(x)$. Considering that an explanation method computes a function $\text{expl}(x, \hat{y})$ of the input $x$ and scalar output prediction $\hat{y}$ (which is one output layer neuron), then a classification explanation $e_{\text{cls}}$ is:

$$\hat{c} = \text{model}_{\text{cls}}(x) \qquad k = \arg\max_i \hat{c}_i \qquad e_{\text{cls}} = \text{expl}(x, \hat{l}_k) \tag{1}$$

A bounding box explanation consists of four different explanations, one for each bounding box component $e_{x_{\min}}, e_{y_{\min}}, e_{x_{\max}}, e_{y_{\max}}$:

$$\hat{x}_{\min}, \hat{y}_{\min}, \hat{x}_{\max}, \hat{y}_{\max} = \text{model}_{\text{bbox}}(x) \qquad e_{x_{\min}} = \text{expl}(x, \hat{x}_{\min}) \qquad e_{y_{\min}} = \text{expl}(x, \hat{y}_{\min}) \tag{2}$$

$$e_{x_{\max}} = \text{expl}(x, \hat{x}_{\max}) \qquad e_{y_{\max}} = \text{expl}(x, \hat{y}_{\max}) \tag{3}$$

In case of explaining the bounding box coordinates, the box offsets predicted by an object detectors are converted to normalized image coordinates before computing the gradient. In case of classification decisions, the logits ($\hat{l}_k$, before softmax probability, $\hat{c} = \text{softmax}(\hat{l})$) are used to compute the gradient. Figure 2 illustrates the explanations generated for each decisions of the cat detection by across detectors. Saliency explanations can be computed for each bounding box of interest in the image.

### 3.2 Multi-object Visualization

In order to summarize the saliency maps of all detections, the individual saliency maps corresponding to each detection are represented using a canonical form. This representation illustrates the most important pixels for the decision explanation. This paper proposes four different methods for combining detection explanations into a single format: principal components, contours, density clustering, and convex polygons. Each method uses a different representation, allowing for detected bounding box, and category to be marked

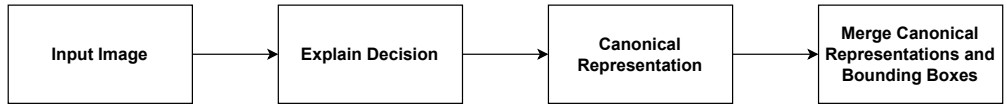

Figure 3: Overview of the Multi-object visualizations pipeline to jointly visualize all detections.

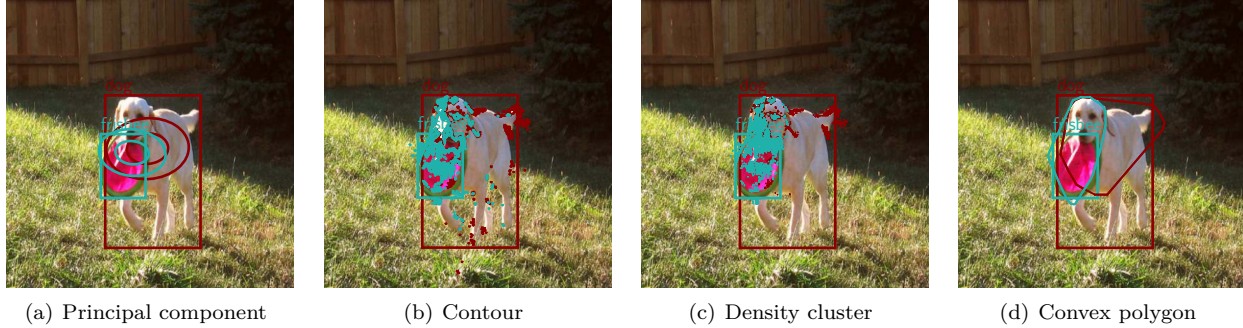

(a) Principal component      (b) Contour      (c) Density cluster      (d) Convex polygon

Figure 4: Multi-object visualizations generated to jointly visualize all detections from EfficientDet-D0 and the corresponding classification explanations generated using SIG in the same color. The combination approach is specified in sub-captions. Explanation pixels are colored same as the corresponding bounding box that is being explained.

using same colors on the input image. The general process is described in Figure 3. An example the four multi-object visualizations are illustrated in Figure 4. Appendix E provides additional details on the multi-object visualization approaches and how different combination methods work. including explanation heatmap samples.

## 4 Experiments

Section 4.1 visually analyzes the explanations generated for different detector and explanation method combinations. Section 4.3 provides the quantitatively evaluates different detector and explanation method combinations. Finally, Section 4.4 estimates an overall ranking for the explanation methods based on user preferences of the explanations produced for each decision. In addition, the multi-object visualization methods are ranked based on user understandability of the detections. In Section F, the procedure to analyze the failures of detector using the proposed approach is discussed.

Most of the experiments use ED0, SSD, and FRN detectors detecting common objects from COCO (Lin et al., 2014). The additional details about these detectors are provided in Table 5. In cases requiring training a detector, different versions of SSD with various pre-trained backbones detecting marine debris provided in Table 6 are used. The marine debris detectors are trained using a train split of the Marine Debris dataset (Valdenegro-Toro, 2019) and explanations are generated for the test split images. These detectors are used only to study how are the explanations change across different backbones and different performance levels (epochs) in Section 4.1.

### 4.1 Visual Analysis

**Across target decision and across detectors**. The saliency maps for the classification and bounding box decisions generated using a particular explanation method for a specific object change across different detectors as shown in Figure 2. All the bounding box explanations of EfficientDet-D0 in certain scenarios provide visual correspondence to the bounding box coordinates.

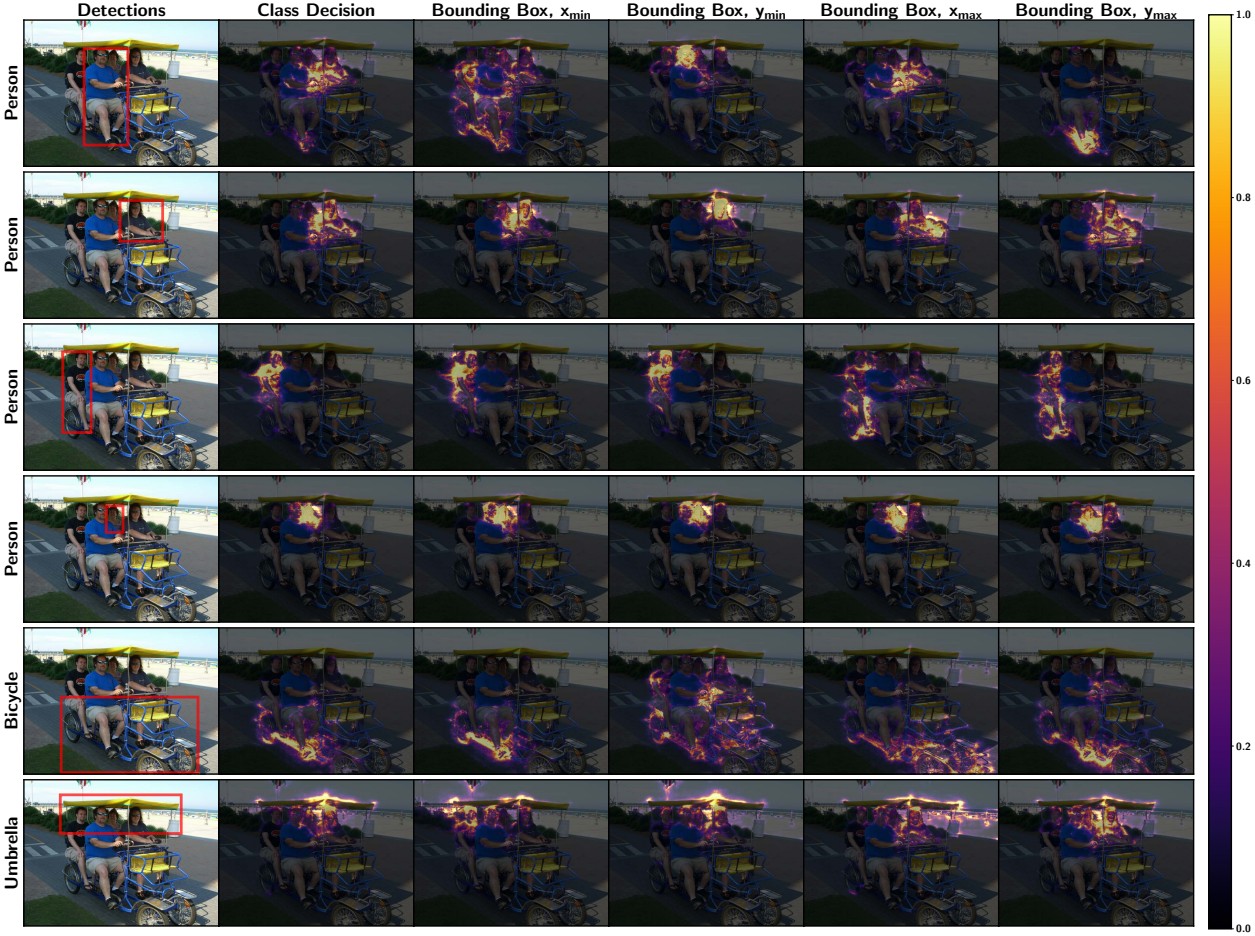

Figure 5: Comparison of classification and bounding box explanations for all detections from EfficientDet-D0 using SIG is provided. Each row provides the detection (red-colored box) followed by the corresponding classification and all bounding box explanation heatmaps.

**Across different target objects**. Figure 5 illustrate that the explanations highlight different regions corresponding to the objects explained. This behavior is consistent in most of the test set examples across the classification and bounding box explanations for all detectors.

Figure 6 illustrates the classification explanations for the wall detection across the 6 different backbones. Apart from the attribution intensity changes, the pixels highlight the different input image pixels, and the saliency map texture changes. MobileNet and VGG16 illustrate thin horizontal lines and highlight other object pixels, respectively. ResNet20 highlights the wall as a thick continuous segment. Figure 18 illustrate the $y_{\min}$ and $y_{\max}$ bounding box coordinate explanations for the chain detection across different backbones. The thin horizontal lines of MobileNet are consistent with the previous example. In addition, VGG16 illustrates a visual correspondence with the $y_{\min}$ and $y_{\max}$ bounding box coordinate by highlighting the upper half and lower half of the bounding box respectively. However, this is not witnessed in other detectors. This behavior is consistent over a set of 10 randomly sampled test set images from the Marine Debris dataset.

The explanations generated using SSD model instances with ResNet20 backbone at different epochs are provided in Figure 7. The model does not provide any final detections at lower epochs. Therefore, the explanations are generated using the target neurons of the output box corresponding to the interest decision in the final detections from the trained model. Figure 7 illustrate variations in the saliency maps starting from a

**Explaining Marine Debris Detections with Different SSD Backbones Using Guided Backpropagation**

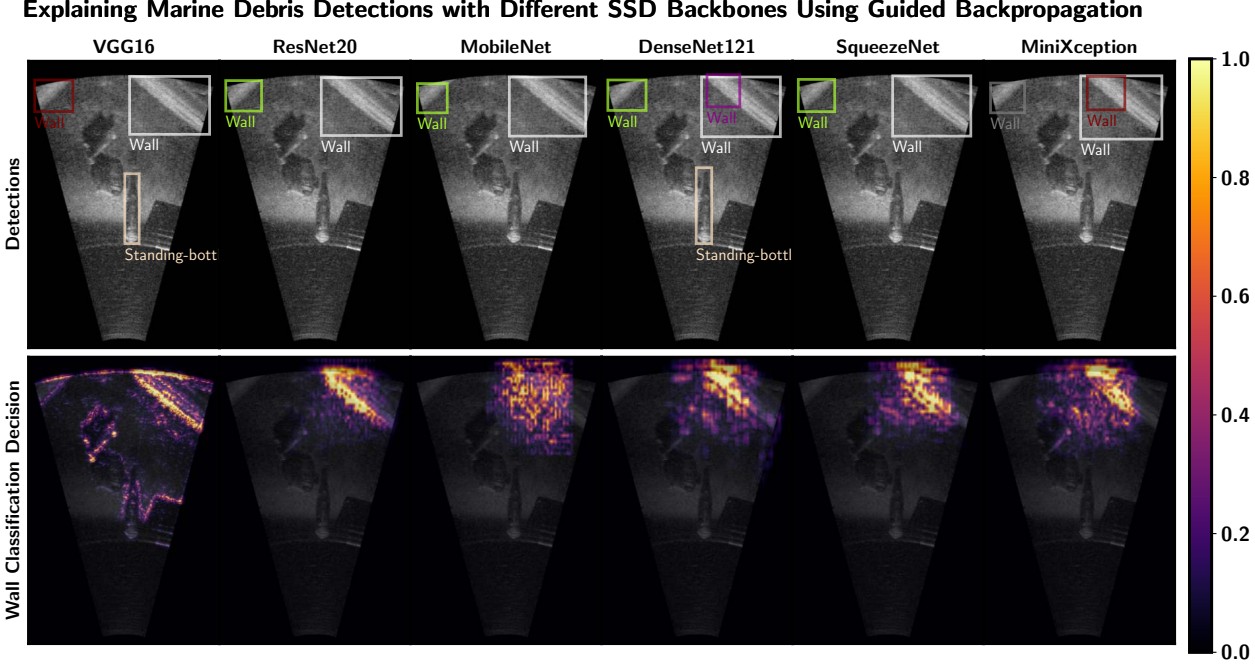

Figure 6: Comparison of class "wall" classification explanations across different SSD backbones. The detections from each SSD backbone are provided in the first row. The wall detection explained is marked using a white-colored box. The explanations vary across each backbone.

randomly initialized model to a completely trained model for the classification decision of the chain detection. The explanations extracted using the random model are dispersed around the features. The explanations slowly concentrate along the chain object detected and capture the object feature to a considerable amount. This behavior is qualitatively analyzed by visualizing the explanation of 10 randomly sampled test set images from the Marine Debris dataset. In the case of the small hook explained in Figure 19, the variations between the random model and the trained model are not as considerable as the previous chain example. This illustrates the variations change with respect to each class.

**SSD-ResNet20 Classification Decision Explanation Using Guided Backpropagation Over Epochs**

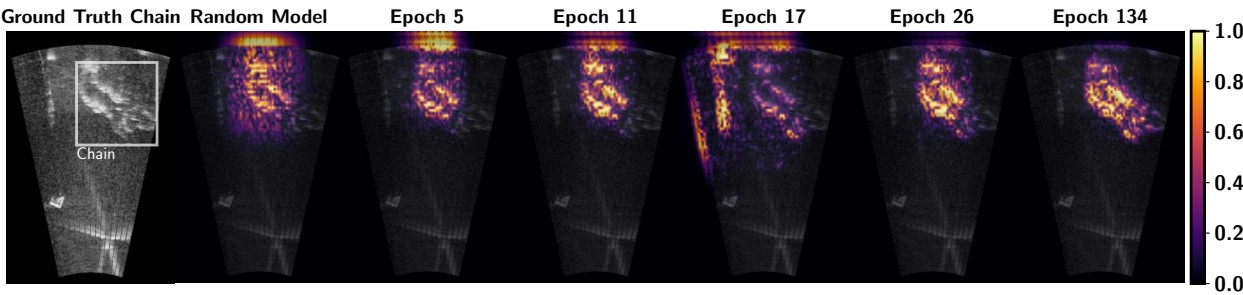

Figure 7: Classification explanation for class "chain" across different epochs (along columns) of SSD-ResNet20 using GBP is illustrated. The first column is the chain ground truth annotation (white-colored box).

## 4.2 Error Analysis

The section analyzes errors made by a detector by generating explanations using the detector explanation approach proposed in this work. The saliency map highlighting the important regions can be used as evidence to understand the reason for the detector failure rather than assuming the possible reasons for detector failure. The failure modes of a detector are wrongly classifying an object, poorly localizing an object, or missing a detection in the image (Petsiuk et al., 2021). As the error analysis study requires ground truth annotations, the PASCAL VOC 2012 images are used. The PASCAL VOC images with labels mapping semantically to COCO labels are only considered as the detectors are trained using the COCO dataset. For instance, the official VOC labels such as sofa and tvmonitor are semantically mapped to couch and tv, respectively, by the model output trained on COCO.

The procedure to analyze a incorrectly classified detection is straightforward. The output bounding box information corresponding to the wrongly classified detection can be analyzed in two ways. The target neuron can be the correct class or the wrongly classified class to generate the saliency maps as shown in Figure 8.

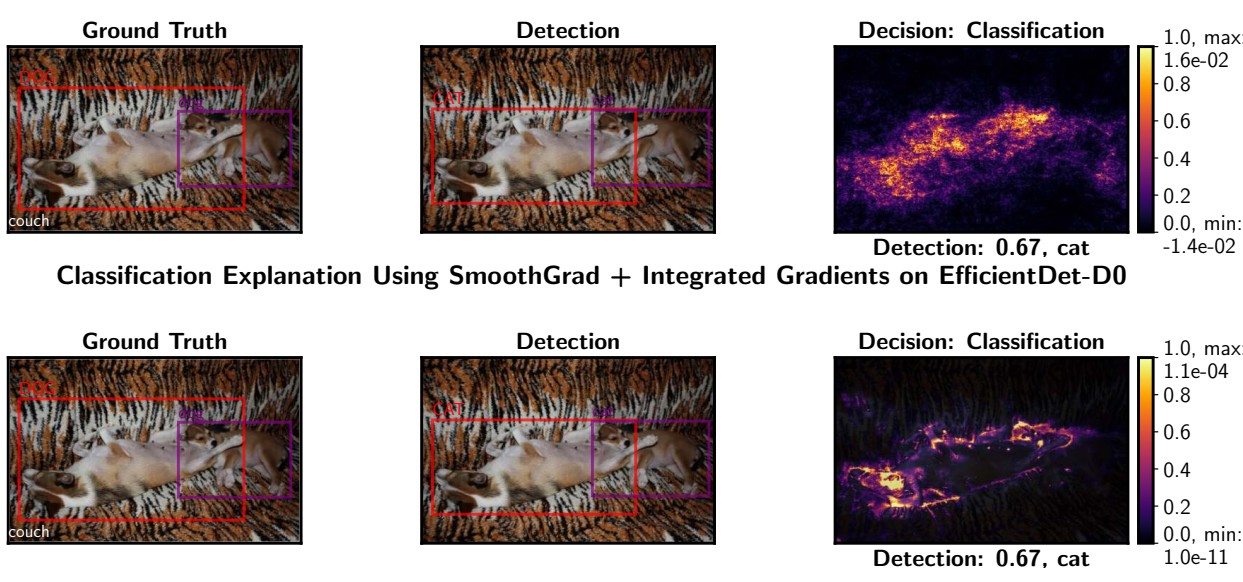

Figure 8: Example error analysis using gradient-based explanations. EfficientDet-D0 wrongly classifies the dog (red-colored box) in ground truth as cat (red-colored box). We display two saliency explanations (GBP and SIG). In this figure, it is clear the model is imagining a long tail for the dog (GBP) and wrongly classifies the dog as a cat. The saliency map highlights certain features of the dog and the background stripes pattern along the edges of the dog body (GBP and SIG). In order to illustrate the tail clearly which is predominant in cats available in COCO dataset, the saliency map is only shown without overlaying on the input image.

More examples of error analysis are available in Section F in the appendix.

## 4.3 Quantitative Evaluation

Evaluating detector explanations quantitatively provides immense understanding on selecting the explanation method suitable for a specific detector. In this section we provide quantitative evaluation of saliency explanations on object detectors.

### 4.3.1 Evaluation Metrics

The quantitative evaluation of the explanations of a detector incorporates causal metrics to evaluate the bounding box and classification explanations. This works by causing a change to the input pixels and measuring the effect of change in model decisions. The evaluation aids in estimating the faithfulness or truthfulness of the explanation to represent the cause of the model decision. The causal metrics discussed in this work are adapted from the previous work (Samek et al., 2021; Petsiuk et al., 2021; 2018). The two variants of causal evaluation metrics based on the cause induced to alter the prediction are deletion and insertion metric. The deletion metric evaluates the saliency map explanation by removing the pixels from the input image and tracking the change in model output. The pixels are removed sequentially in the order of the most important pixels starting with a larger attribution value and the output probability of the predicted class is measured. The insertion metric works complementary to the deletion metric by sequentially adding the most important pixel to the image and causing the model decision to change. Using deletion metric, the explanation methods can be compared by plotting the fraction of pixels removed along $x$-axis and the predicted class probability along $y$-axis. The method with lower Area Under the Curve (AUC) illustrates a sharp drop in probability for lesser pixel removal. This signifies the explanation method can find the most important pixels that can cause a significant change in model behavior. The explanation method with less AUC is better. In the case of insertion metric, the predicted class probability increases as the most relevant pixels are inserted. Therefore, an explanation method with a higher AUC is relatively better. Petsiuk et al. (2021) utilize constant gray replacing pixel values and blurred image as the start image for deletion and insertion metric calculation respectively.

**Effects Tracked**. The previous work evaluating the explanations of detector decisions utilize insertion and deletion metric to track the change in the bounding box IoU and classification probability together. Petsiuk et al. (2021) formulate a vector representation involving the box coordinates, class, and probability. The similarity score between the non-manipulated and manipulated vectors are tracked. However, this work performs an extensive comparison of explanation methods for each decision of a detector by tracking the change in maximum probability of the predicted class, Intersection over Union (IoU), distance moved by the bounding box (in pixels), change in height of the bounding box (in pixels), change in width of the bounding box (in pixels), change in top-left $x$ coordinate of the bounding box (in pixels), and change in top-left $y$ coordinate of the bounding box (in pixels). The box movement is the total movement in left-top and right-bottom coordinates represented as euclidean distance in pixels. The coordinates distances are computed using the interest box corresponding to the current manipulated image and the interest box corresponding to the non-manipulated image. This extensive evaluation illustrates a few explanation methods are more suitable to explain a particular decision. As declared in the previous sections, the image origin is at the top-left corner. Therefore, a total of 7 effects are tracked for each causal evaluation metric.

**Evaluation Settings**. The previous section establishes the causal deletion and insertion metric along with the 7 different effects. In this section, two different settings used to evaluate the detectors using the causal metrics are discussed.

*Single-box Evaluation Setting.* The detector output changes drastically on manipulating the input image at different fractions. The detector output box detecting the object in the non-manipulated input image is termed the principal box. In this setting, the 7 effects of the principal box are tracked across insertion and deletion of input pixels. This aids in capturing how well the explanation captures the true causes of the principal box prediction. Therefore, the impact of the explanation on the actual detected box is analyzed. The effects measured for the single-box setting are bounded because the value of the principal box is always measurable. This is called a single-box setting because only the changes in the principal box are tracked. For instance, the output principal box tracked will always output a probability or bounding box coordinate. This probability or bounding box coordinate can be used to track the effects for all manipulated input images.

*Realistic Evaluation Setting.* In this evaluation setting, all 7 effects are tracked for the complete object detector output involving the post-processing steps of a detector. The faithfulness of the explanation to the detection pipeline is analyzed. Therefore, how well the explanation depicts the cause of the detector decision is evaluated. In this setting, the current detection for a particular manipulated input image is matched to the interest detection by checking the same class and an IoU threshold greater than 0.9. For various manipulated

| Cause | Effect Tracked | Evaluation Setting |
|---|---|---|
| ↓ Deletion (**D**) | Class Maximum Probability (**C**) | Single-box (**S**) |
| ↑ Insertion (**I**) | Box IoU (**B**) | Realistic (**R**) |
| | Box Movement Distance (**M**) | |
| | Box X-top (**X**), Box Y-top (**Y**) | |
| | Box Width (**W**), Box Height(**H**) | |

Table 1: The components of an evaluation metric with the respective tag for each component is provided in bracket. Therefore, a total of 28 metrics ($2 \times 7 \times 2$) are used in the work. The abbreviation for each evaluation metric is read from the left-side. For instance, DCS means Deletion metric tracking the change in maximum probability of the output box chosen in single-box setting.

input images, there is no current detection matching the interest detection. Therefore, depending on the effect tracked and to calculate AUC, a suitable value is assigned to measure the effect. For instance, if the effect tracked is the class probability for deletion metric and none of the current detection matches with the interest detection, a zero class probability is assigned. Similarly, if the effect tracked is box movement in pixels for deletion metric, the error in pixels increases to a large value. In such scenarios, instead of assigning a large value, assuming the 4 coordinates of interest detection are at a particular image corner and the 4 box coordinates of the final detection are in the opposite corner, the maximum distance is assigned to the box movement in pixels.

**Interpretation Through Curves**. Given the different causes induced to change model output, effects tracked, and evaluation setting for the detector, 28 causal evaluation metrics are used in this work. Table 1 summarize the combinations of different aspects of the evaluation metrics with short-form tags.

To interpret a causal evaluation metric, a graph is drawn tracking the change of the effect tracked along the $y$-axis and the fraction of pixels manipulated along the $x$-axis. For instance, consider the scenario of deleting image pixels sequentially to track the maximum probability of the predicted class at single-box evaluation setting. The $x$-axis is the fraction of pixels deleted. The $y$-axis is the maximum probability of the predicted class at the output of the box tracked. In this work, the curve drawn is named after the combination of the causal evaluation metrics, effects tracked, end evaluation settings. The curves are the DCS curve, DBS curve, ICS curve. For instance, the DCS curve is the change in the maximum probability for the predicted class (C) at the single output box (S) due to removing pixels (D). The curves are the evaluation metrics used in this work and also called as DCS evaluation metric (deletion + class maximum probability + single-box setting), DBS (deletion + box IoU + single-box setting) evaluation metric, and so on.

In order to compare the performance of explanation methods to explain a single detection, as stated before, the AUC of a particular evaluation metric curve is estimated. The corresponding AUC is represented as $\text{AUC}_{<\text{evaluation\_metric\_name}>}$. In order to estimate a global metric to compare the explanation methods explaining a particular decision of a detector, the average AUC, represented as $\text{AAUC}_{<\text{evaluation\_metric\_name}>}$, is computed. As the explanations are provided for each detection, the evaluation set is given by the total number of detections. The total detections in the evaluation set are the sum of detections in each image of the evaluation set. The average evaluation metric curve is computed by averaging the evaluation metric curve at each fraction of pixels manipulated across all detections. AAUC of a particular evaluation metric curve is the AUC of the average evaluation metric curve.

### 4.3.2 Results

Figure 9 illustrates the AAUC computed by evaluating the explanations of each bounding box coordinate is similar across different evaluation metrics curves. This similarity is consistent for all the detectors and explanation methods combinations evaluated. Therefore, the explanation methods quantitatively explain each bounding box coordinate decisions with similar performance. In this work, the AAUC for the bounding box decision is computed by averaging the AUC of all the evaluation metric curves corresponding to all

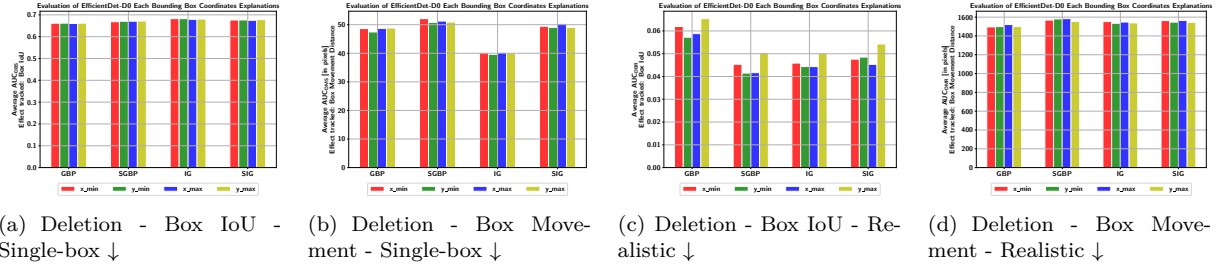

(a) Deletion - Box IoU - Single-box ↓

(b) Deletion - Box Movement - Single-box ↓

(c) Deletion - Box IoU - Realistic ↓

(d) Deletion - Box Movement - Realistic ↓

Figure 9: The figure illustrates the average AUC, AAUC, for the evaluation metric curves obtained by tracking box IoU (a, c) and box movement distance (b, d) as the pixels are deleted sequentially. Each bar corresponds to the AAUC estimated by evaluating explanations generated for each bounding box coordinate decisions using the explanation methods specified in the $x$-axis of all detection made by EfficientDet-D0 in the evaluation set images. AAUC is computed by averaging the AUC of all the evaluation metric curves generated using the combination specified in the sub-captions. Lower AAUC is better in all the plots.

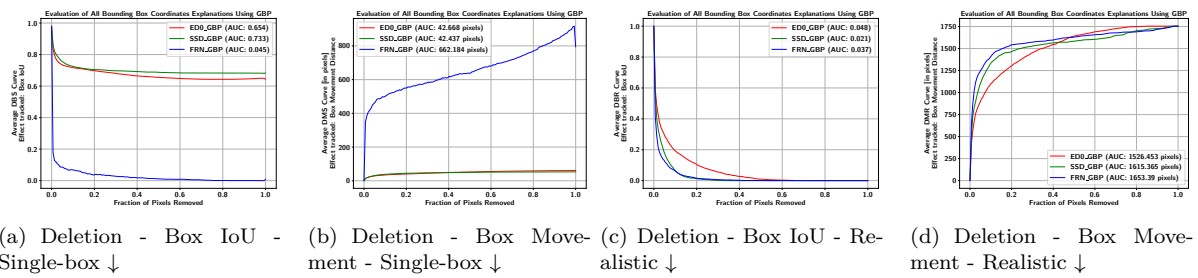

(a) Deletion - Box IoU - Single-box ↓

(b) Deletion - Box Movement - Single-box ↓

(c) Deletion - Box IoU - Realistic ↓

(d) Deletion - Box Movement - Realistic ↓

Figure 10: Comparison of average curves obtained by tracking box IoU (a, c) and box movement distance (b, d) as the pixels are deleted sequentially. Each average curve is the average of the evaluation curves plotted by evaluating the explanations of all bounding box coordinate decisions across all the detections by the respective detector. The explanations are generated using GBP. The evaluation metric curve is generated using the combination specified in the sub-captions.

the box coordinate explanations. This offers the means to evaluate the explanation methods across all the bounding box coordinate decisions.

Figure 10 and Figure 11 illustrate quantitatively complementary trends in the evaluation metric curves plotted by tracking box movement distance in pixels and box IoU. The IoU decreases and box movement distance increases as the pixels are deleted sequentially as shown in Figure 10. Similarly, Figure 11 illustrates the increase in box IoU and decrease in box movement distance as pixels are inserted to a blurred version of the image. There is a large difference in the AAUC between the single-stage and two-stage detectors. This is primarily due to the RPN in the two-stage detectors. The proposals from RPN are relatively more sensitive to the box coordinate change than the predefined anchors of the single-stage detectors. In addition, Figure 10(d) and Figure 11(d) indicates the steady change of box coordinates in the final detections of the EfficientDet-D0. However, SSD and Faster R-CNN saturate relatively sooner. In the remainder of this work, the ability of the box IoU effect is used for quantitative evaluation. This is only because the box IoU effect offers the same scale between 0 to 1 as the class maximum probability effect. In addition, both box IoU and class maximum probability effect follow the trend lower AUC is better for the deletion case. However, it is recommended to consider all the box IoU and box movement distance effects at the level of each box coordinate for a more accurate evaluation.

Figure 12 and Figure 17 aids in understanding the explanation method interpreting both the classification and bounding box decision of a particular detector more faithful than other explanation methods. Figure 12(a) illustrate SSD512 classification decisions are better explained by SGBP at single-box setting for deletion metrics. However, the bounding box decisions are not explained as well as the classification decisions. Figure

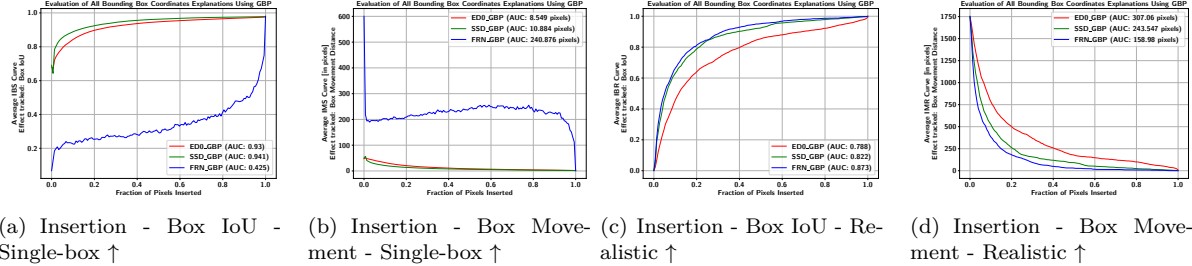

(a) Insertion - Box IoU - Single-box ↑

(b) Insertion - Box Movement - Single-box ↑

(c) Insertion - Box IoU - Realistic ↑

(d) Insertion - Box Movement - Realistic ↑

Figure 11: Comparison of average curves obtained by tracking box IoU (a, c) and box movement distance (b, d) as the pixels are inserted sequentially. Each average curve is the average of the evaluation curves plotted by evaluating the explanations of all bounding box coordinate decisions across all the detections by the respective detector. The explanations are generated using GBP. The evaluation metric curve is generated using the combination specified in the sub-captions.

12(b) illustrate a similar scenario for SGBP with EfficientDet-D0 and Faster R-CNN at the realistic setting for deletion metrics. However, all selected explanation methods explain the bounding box and classification decisions of SSD512 relatively better at the single-box setting for insertion metrics. In general, none of the selected explanation methods explain both the classification and bounding box regression decisions substantially well compared to other methods for all detectors. This answers EQ13. Similarly, none of the detectors is explained more faithfully for both classification and bounding box decisions among the selected detectors by a single method across all evaluation metrics discussed. This is illustrated by no explanation methods (by different colors) or no detectors (by different alphabets) being represent in the lower left rectangle or upper right rectangle in Figure 12 and Figure 17 respectively.

Figure 14(a) and Figure 14(c) illustrate AAUC of the classification saliency maps and the saliency maps combined using different merging methods are different in certain scenarios while tracking the maximum probability. The AAUC of all the box coordinate saliency maps is provided for a baseline comparison. This denotes the effect on maximum probability by removing pixels in the order of most important depending on the all box coordinates saliency maps. Similarly, Figure 14(b) and Figure 14(d) illustrate the similarity in the AAUC of all box coordinate explanations and the merged saliency maps while tracking the box IoU. In Figure 14(a), the evaluation of the GBP classification saliency map is less faithful than the merged saliency map. Therefore, the merged saliency map represents the classification decision more faithfully than the standalone classification explanation in the case of EfficientDet-D0. However, Figure 14(a) and Figure 14(c) illustrate in the case of SGBP explaining EfficientDet-D0 and certain cases of Faster R-CNN respectively separately classification saliency maps are more faithful in depicting the classification decision. The larger AAUC for all the box coordinate saliency maps generated using each method for Faster R-CNN indicate the box saliency maps are not faithful to the bounding box decisions of Faster R-CNN. This is coherent with the visual analysis. Therefore, in certain scenarios merging is helpful to represent the reason for a particular decision. However, each individual saliency map provides peculiar information about the detection. For instance, the visual correspondence shown in Figure 2 to each bounding coordinate information is seen only at the level of individual box coordinate explanations.

An overall comparison of all quantitative metrics is shown in Figure 13. For the purpose of understanding, the ranking of detectors better explained by a particular explanation method is provided in Table 2. The ranking of explanation methods explaining a particular detector is provided in Table 3. SGBP performs relatively better across all selected detectors. In addition, IG is ranked least across all the selected detectors. SSD detector is better explained by all the explanation methods. One of the reasons can be SSD is a simpler architecture compared to EfficientDet-D0 and Faster R-CNN. EfficientDet-D0 and Faster R-CNN include a Bi-directional Feature Pyramid Network (BiFPN) and Region Proposal Network (RPN) respectively. However, further experiments should be conducted for validation.

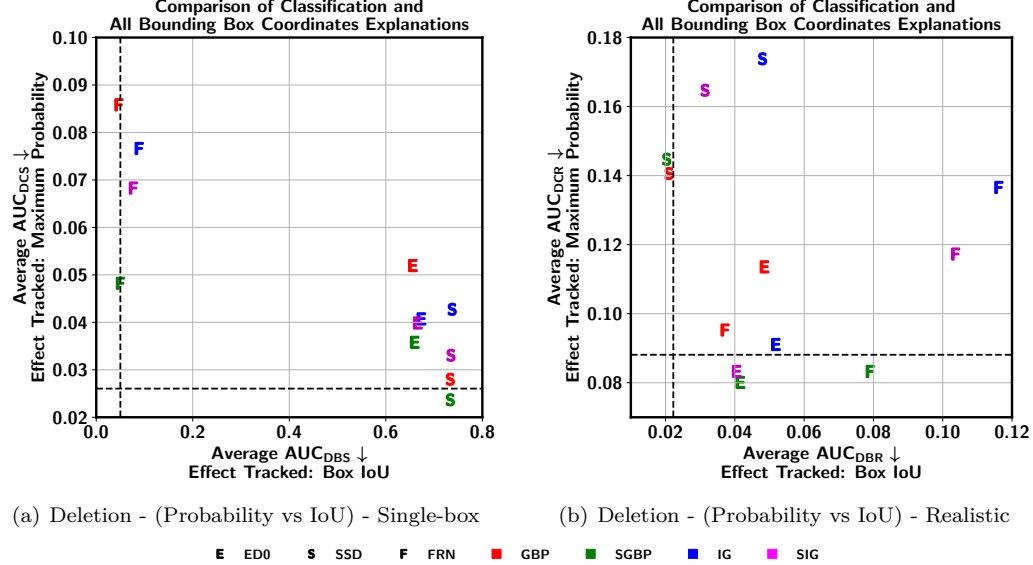

(a) Deletion - (Probability vs IoU) - Single-box    (b) Deletion - (Probability vs IoU) - Realistic

Figure 12: Comparison between the Deletion AAUC of the evaluation metric curves for the classification and all bounding box coordinate explanations generated using different explanation methods across all detectors. This offers a means to understand the explanation method generating more faithful explanations for both classification explanations and all bounding box coordinates. As the curves to compute the respective AUC are computed using deletion metric, lower values in both axis are better. The explanation methods (highlighted with different colors) placed at a lower value in the $x$-axis and $y$-axis perform relatively better at explaining the box coordinates and classification decisions respectively. The detectors (marked with different alphabets) placed at a lower value in $x$-axis and $y$-axis are relatively better explained for the box coordinates and classification decisions respectively.

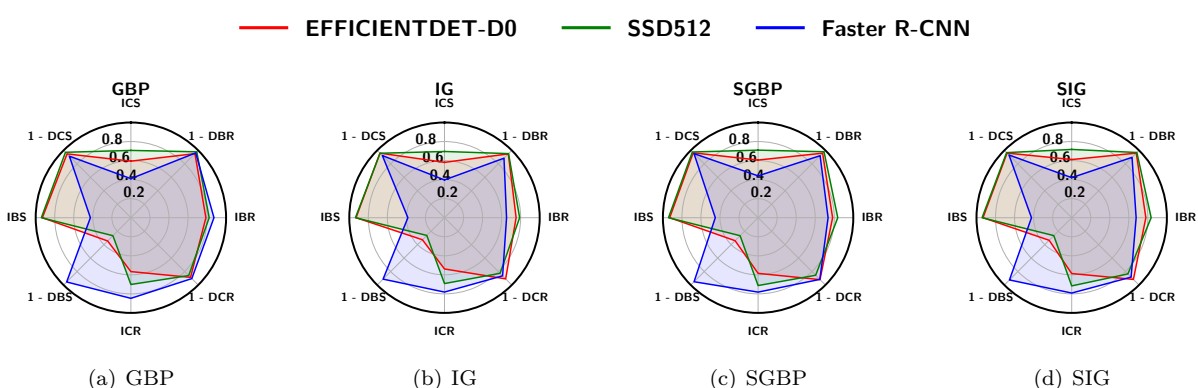

(a) GBP          (b) IG          (c) SGBP          (d) SIG

Figure 13: Multi-metric comparison of quantitative results. According to these metrics, all methods perform similarly when considering all object detectors. The user study and visual inspection of explanation heatmaps reveal more information.

## 4.4 Human-centric Evaluation

The human-centric evaluation ranks the explanation methods for each detector and ranks the multi-object visualization methods with a user study. All important details of the user study are presented in Appendix G.

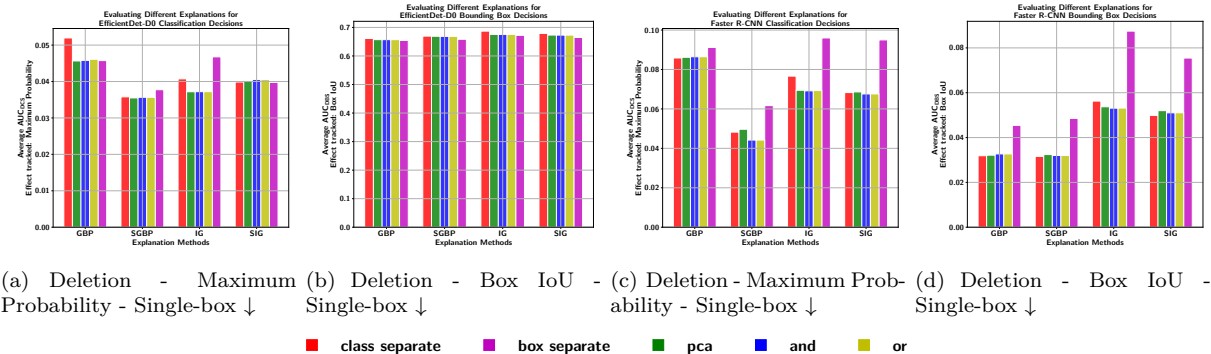

(a) Deletion - Maximum Probability - Single-box ↓  (b) Deletion - Box IoU - Single-box ↓  (c) Deletion - Maximum Probability - Single-box ↓  (d) Deletion - Box IoU - Single-box ↓

Figure 14: Comparison of average AUC, AAUC, for the evaluation metric curves obtained by tracking maximum probability (a, c) and box IoU (b, d) as the most important pixels based on the explanation generated using the explanation methods specified in the $x$-axis are deleted sequentially. All the explanations are generated for detection made by EfficientDet-D0 (left) and Faster R-CNN (right) in the evaluation set images. Lower AAUC is better in both plots.

| IM | OD | DCS | ICS | DBS | IBS | DCR | ICR | DBR | IBR | Overall Rank |
|----|----|-----|-----|-----|-----|-----|-----|-----|-----|--------------|
| GBP | ED0 | 2 | 2 | 2 | 2 | 2 | 3 | 3 | 3 | 3 |
|  | SSD | 1 | 1 | 3 | 1 | 3 | 2 | 1 | 2 | 1 |
|  | FRN | 3 | 3 | 1 | 3 | 1 | 1 | 2 | 1 | 2 |
| SGBP | ED0 | 2 | 2 | 2 | 2 | 1 | 3 | 2 | 2 | 2 |
|  | SSD | 1 | 1 | 3 | 1 | 3 | 2 | 1 | 1 | 1 |
|  | FRN | 3 | 3 | 1 | 3 | 2 | 1 | 3 | 3 | 3 |
| IG | ED0 | 1 | 2 | 2 | 2 | 1 | 3 | 2 | 2 | 2 |
|  | SSD | 2 | 1 | 3 | 1 | 3 | 2 | 1 | 1 | 1 |
|  | FRN | 3 | 3 | 1 | 3 | 2 | 1 | 3 | 3 | 3 |
| SIG | ED0 | 2 | 2 | 2 | 2 | 1 | 3 | 2 | 2 | 2 |
|  | SSD | 1 | 1 | 3 | 1 | 3 | 2 | 1 | 1 | 1 |
|  | FRN | 3 | 3 | 1 | 3 | 2 | 1 | 3 | 3 | 3 |

Table 2: Ranking of all detectors for a particular explanation method based on the quantitative evaluation metrics. A lower value is a better rank. The detector better explained by a particular explanation method is awarded a better rank. Each detector is ranked with respect to each evaluation metric considering a particular explanation method. The column names other than the last column and the first two columns represent the AAUC for the respective evaluation metric. The overall rank is computed by calculating the sum along the row and awarding the best rank to the lowest sum. OD - Object detectors, IM - Interpretation method.

### 4.4.1 Rank Generation

**Ranking Explanation Methods**. Previous work assess the user trust in the model explanations generated by a particular explanation method (Petsiuk et al., 2021; Selvaraju et al., 2020; Ribeiro et al., 2016). As user trust is difficult to evaluate precisely. This work in contrast to the the previous works actually estimate the user preferability of the explanation methods. The user preferability for the methods GBP, SGBP, IG, and SIG are evaluated by comparing two explanations corresponding to a particular predictions. In this study, the explanation methods are compared directly for a particular interest detection and interest decision across SSD, EDO, and FRN detector separately. The evaluation identifies the relatively more trusted explanation method by the users for a particular detector. The explanation methods are ranked by relatively rating the

| OD | IM | DCS | ICS | DBS | IBS | DCR | ICR | DBR | IBR | Overall Rank |
|----|-----|-----|-----|-----|-----|-----|-----|-----|-----|--------------|
| ED0 | GBP | 4 | 3 | 1 | 2 | 4 | 3 | 3 | 1 | 3 |
| | SGBP | 1 | 2 | 2 | 4 | 1 | 2 | 2 | 2 | 2 |
| | IG | 3 | 4 | 4 | 3 | 3 | 4 | 4 | 4 | 4 |
| | SIG | 2 | 1 | 3 | 1 | 2 | 1 | 1 | 3 | 1 |
| SSD | GBP | 2 | 3 | 2 | 3 | 1 | 3 | 2 | 3 | 3 |
| | SGBP | 1 | 2 | 1 | 2 | 2 | 2 | 1 | 1 | 1 |
| | IG | 4 | 4 | 4 | 4 | 4 | 4 | 7 | 4 | 4 |
| | SIG | 3 | 1 | 3 | 1 | 3 | 1 | 3 | 2 | 2 |
| FRN | GBP | 4 | 3 | 1 | 2 | 2 | 1 | 1 | 1 | 1 |
| | SGBP | 1 | 1 | 2 | 1 | 1 | 3 | 2 | 2 | 2 |
| | IG | 3 | 4 | 4 | 4 | 4 | 4 | 4 | 4 | 4 |
| | SIG | 2 | 2 | 3 | 3 | 3 | 2 | 3 | 3 | 3 |

Table 3: Ranking of all the explanation methods for a particular detector based on the quantitative evaluation metrics. A lower value is a better rank. The explanation method better explaining a particular detector is awarded a better rank. Each detector is ranked with respect to each evaluation metric considering a particular explanation method. The column names other than the last column and the first two columns represent the average AUC for the respective evaluation metric. The overall rank is computed by calculating the sum along the row and awarding the best rank to the lowest sum. OD - Object detectors, IM - Interpretation method.

explanations generated using different explanation methods for a particular detection made by a detector. The rating serves as a measure of user preference.

A pair of explanations generated by different explanation methods using the same interest decision and same interest detection for the same detector is shown to a number of human users as shown in Figure 38. The detector, interest decision, interest detection, and explanation method used to generate explanations are randomly sampled for each question and each user. In addition, the image chosen for a particular question is randomly sampled from an evaluation set. The evaluation set is a randomly sampled set containing 50 images from the COCO test 2017. This avoids incorporating any bias into the question generation procedure. Each question is generated on the fly for each user performing the task. The explanations are named Robot A explanation and Robot B explanation to conceal the names of the explanation methods to the user. The robots are not detectors. In this study, the robots are treated as explanation methods. Robot A explanation and Robot B explanation for each question is randomly assigned with a pair of explanation method output. This is done to reduce the bias due to positioning and ordering bias of the explanations as shown to users. The task provided for the user is to rate the quality of the Robot A explanation based on the Robot B explanation. The available options are provided in Table 4.

| Options | A Score | B Score |
|---------|---------|---------|
| Robot A explanation is much better | 2 | -2 |
| Robot A explanation is slightly better | 1 | -1 |
| Both explanations are same | 0 | 0 |
| Robot A explanation is slightly worse | -1 | 1 |
| Robot A explanation is much worse | -2 | 2 |

Table 4: User study options and scores awarded to respective explanations.

A single question in the evaluation is treated as a game between two randomly matched players. The explanation methods are the players. The game result depends on the explanation quality produced by the competing explanation methods for a particular detection decision. In case of a draw, both explanation

methods receive the same score. During non-draw situations, the points won by a particular explanation method are the points lost by the other explanation method. By treating all the questions answered by numerous users as individual games, the global ranking is obtained using the Elo rating system (Elo, 1978). Each explanation method is awarded an initial Elo rating of 1000.

**Ranking Multi-Object Visualization Methods**. The rank for multi-object visualization methods is obtained by voting for the method producing the most understandable explanation among the four methods. Each user is asked a set of questions showing the multi-object visualization generated by all four methods. The user is provided with a *None of the methods* option to chose during scenarios where all the multi-object visualizations generated are confusing and incomprehensible to the user. The methods are ranked by counting the total number of votes each method has obtained. The experiment is performed using COCO 2017 test split and the VOC 2012.

### 4.4.2   Results

Each user is requested to answer 10 questions. The total number of questions is split as 7 and 3 between Task 1 and Task 2 respectively. 52 participants have answered the user study for both task 1 and task 2. The participants range across researchers, students, deep learning engineers, office secretaries, and software engineers.

Figure 15 indicates SGBP provide relatively more reasonable explanations with higher user preferability for both single-stage detectors. Similarly, SIG is preferred for the two-stage detector.

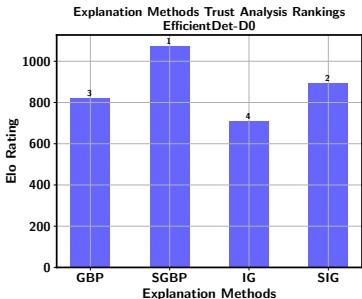 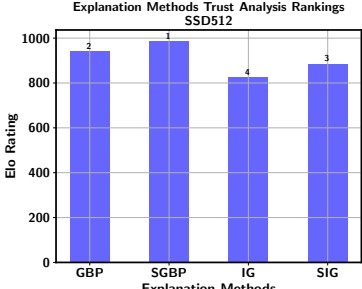 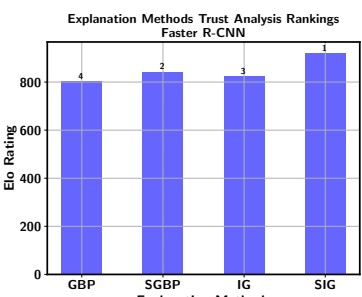

Figure 15:   Ranking obtained for the explanation methods from the user trust study for each detector selected in this work. An initial Elo rating of 1000 is used for all explanation methods. The explanation method with a higher Elo rating has gained relatively more user preferability in the random pair-wise comparisons of explanations for each detector. The rank of a particular method is provided on the top of the bar corresponding to the method.

Figure 16(a) illustrates the top two ranks are obtained by SmoothGrad versions of the SGBP and IG for all detectors. GBP relatively performs in the middle rank in the majority of cases. SGBP achieves the first rank in both the human-centric evaluation and functional evaluation. Figure 16(a) illustrates the overall ranking taking into account all the bounding box and classification explanations together. The ranking is similar in analyzing the bounding box and classification explanations separately.

With the ranking of multi-object visualization methods, it is clearly evident that majority of the users are able to understand convex polygon-based explanations. 18 answers among the total 156 are *None of the methods* because none of the four other multi-object visualization methods provided a legible summary of all the explanation methods and detections. The users have selected principal component-based visualization in cases involving less than 3 detections in an image. In addition, *None of the methods* is chosen in most of the cases involving more than 9 detections or more than 3 overlapping detections in an image. Among the total participants, only 89 users (57%) agree with the convex polygon-based visualization. Therefore, by considering the remaining 43% users, there is a lot of need to improve the multi-object visualization methods discussed in this work and achieve a better summary.

# 5 Conclusions and Future Work

Explaining convolutional object detectors is crucial given the ubiquity of detectors in the fields of autonomous driving, healthcare, and robotics. In this paper we extend post-hoc gradient-based explanation methods to explain both classification and bounding box decisions of EfficientDet-D0, SSD512, and Faster R-CNN.

Additionally, in order to integrate explanations for all detected bounding boxes into a single output images, we propose four multi-object visualization methods to merge explanations of a particular decision, namely PCA, Contours, Density clustering, and Convex polygons.

We evaluate these detectors and their explanations using a set of quantitative metrics (insertion and deletion of pixels according to saliency map importance, with single-box and realistic settings) and with a user study to understand how useful these explanations are to humans.

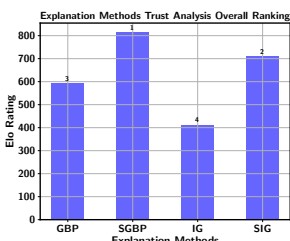

(a) Explanation Methods

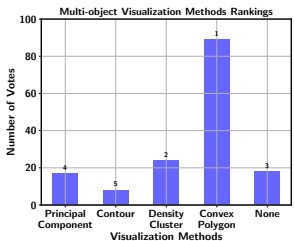

(b) Multi-Object Visualization Methods

Figure 16: Ranking obtained from the user study considering all user answers. The rank of a particular method is provided on the top of the bar corresponding to the method.

Insertion and deletion metrics indicate that SGBP provides more faithful explanations in the overall ranking. In general there is no detector that clearly provides better explanations, as a best depends on the criteria being used, but visual inspection indicates a weak relationship that newer detectors (like EfficientDet) have better explanations without artifacts (as shown in Figure 2), and that different backbones do have an influence on the saliency map quality (Figure 6).

The user study reveals a human preference for SGBP explanations for SSD and EfficientDet (and SIG for Faster R-CNN), which is consistent with the quantitative evaluation, and for multi-object explanation visualizations, convex polygons are clearly preferred by humans.

In addition, we analyze some failure modes of a detector using the formulated explanation approach and provide several examples. The overall message of our work is to always explain both object classification and bounding box decisions, and that it is possible to combine explanations into a single output image through convex polygon representation of the saliency map.

Finally, we developed an open-source toolkit, DExT, to explain decisions made by a detector using gradient-based saliency maps, to generate multi-object visualizations, and to analyze failure modes.

We expect that DExT and our evaluation will contribute to the development of holistic explanation methods for object detectors, considering all their output bounding boxes, and both object classification and bounding box decisions.

**Limitations**. Our work encompasses the following limitations. The first is about the pixel insertion/deletion metrics might be difficult to interpret (Grabska-Barwinska et al., 2021) and more advanced metrics could be used (Tomsett et al., 2020). However, the metric selected should consider the specifics of object detection. Firstly, both classification and bounding box regression should be evaluated with the selected metric. Moreover, as detectors are prone to non-local effects, removing pixels from the image (Rosenfeld et al., 2018) can cause bounding boxes to appear or disappear. Therefore, special tracking of a particular box is needed. In our work, we extend the classic pixel insertion/deletion metrics (Ancona et al., 2019) for object detection considering these two aspects.

The second limitation is about the user study. Given the challenges in formulating a bias-free question, in our user study we ask users to select which explanation method is better. This is a subjective human judgment and does not necessarily have to correspond with the true input feature attribution made by the explanation

method. Another part of the user study is comparing multi-object visualization methods, where we believe there is a much clearer conclusion. The novelty of our work is to combine quantitative, qualitative, and a user study, to empirically evaluate saliency explanations for object detectors considering object classification and bounding box regression decisions.

In general, saliency methods are prone to heavy criticisms questioning the reliability of the methods. This study extends a few gradient-based saliency methods for detectors and conducts extensive evaluation. However, we acknowledge that there are other prominent saliency methods to study.

Our work evaluates and explains real-world object detectors without any toy example. The literature has previously performed basic sanity checks on toy usecases that does not include multiple localization and classification outputs. In addition, object detectors are categorized on the basis of number of stages (single-stage Liu et al. (2016); Tan et al. (2020) and two-stage Ren et al. (2017)), availability of anchors (anchor-based Liu et al. (2016); Tan et al. (2020) and anchor-free Redmon et al. (2016); Tian et al. (2019)), and vision transformer based detectors Carion et al. (2020); Beal et al. (2020). In this work, we explain detectors specific to certain groups (SSD512, Faster R-CNN, and EfficientDet) and leave anchor-free and transformer-based detectors for future work, as they are not trivial to explain using gradient-based saliency method.

**Broader Impact Statement**. As concerns on AI safety is increasing, explainable machine learning is imperative to gain human trust and satisfy the legal requirements. Any machine learning model user for human applications should be able to explain its predictions, in order to be audited, and to decide if the predictions are useful or further human processing is needed. Similarly, such explanations are pivotal to earn user trust, increase applicability, address safety concerns for complex object detection models.

We expect that our work can improve the explainability of object detectors, by first steering the community to explain all object detector decisions (bounding box and object classification), considering to visualize all saliency explanations in a single image per detector decision, and to evaluate the non-local effect of image pixels into particular detections. We believe that saliency methods can be used to partially debug object detection models. Consequently, saliency methods are useful to explain detector and address the trustworthiness and safety concerns in critical applications using detectors.

However, additional validation of explanations is needed. We also perform sanity checks in object detectors [reference withheld due to double blind submission] with similar conclusions and validation of saliency map quality. Additional large scale user studies could be done to evaluate how useful these explanations are for humans, instead of just asking which explanation method is better.

Even though fully white-box interpretable models would be the best solution (Rudin, 2019), this is not yet available at the model scale required for high object detection performance.

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
