# OpenReview forum: "DExT: Detector Explanation Toolkit"
_TMLR — Rejected by TMLR_

### Review · Reviewer_UFGg · 2023-01-16

**Summary Of Contributions:**

The paper focuses on the problem of producing explanations for object detectors. Explanations that are robust and human understandable are crucial to increase the trust of users in these models in safety critical applications, such as autonomous driving and medical imaging. Building on a growing literature for explanation methods, the authors propose a methodology to provide saliency maps for both the bounding boxes and the prediction of the model, visualize explanations for multiple objects in a single output, and collect all this into a toolkit.

The contributions of the paper are the following:
* As mentioned above, a software package for producing explanations, evaluating them and identifying errors. The code is provided with the paper.
* Extending gradient based explanations to bounding boxes and predictions of object detectors
* Merge all detection outputs in a single explanation, with a multi-object visualization procedure. The authors propose 4 approaches to operate this merging: principal components (drawing ellipses centered at the mass center depicting the minimum and maximum spread of saliency maps densities), contours, density clustering and convex polygons (covering the density clustered saliency map pixels).
* An evaluation procedure, including both qualitative and quantitative analysis: the analysis consider all possible combinations of selected detectors and saliency map producing approaches (see details in the next paragraph). For the quantitative analysis, the authors build on previously introduced pixel removal or insertion based metrics, and identify 7 effects to track for a detector:  maximum probability of the predicted class, Intersection over Union (IoU), distance moved by the bounding box, change in height of the bounding box, change in width of the boundingbox, change in top-left coordinate of the bounding box in both directions. The 5 last effects are measured in pixels. Finally, they conducted a user study, assessing the preference of users among the tested approaches (detector x explanation x merging procedure).
* A brief description of failure detection in the appendix.

In order to conduct their study, the authors select 3 detectors: SSD512, Faster R-CNN and EfficientDet-D0. These detectors use feature extraction backbones from different architectural families: VGG, Resnet and EfficientNet. They also select 4 gradient-based approaches to produce saliency maps: guided backpropagation, integrated gradients, and the two approaches combined with SmoothGrad. They present results on 2 datasets: COCO and Marine Debris.

**Audience:**

Yes

**Broader Impact Concerns:**

Given the motivation of the work, targeting to increase trustworthiness of detection methods for safety-critical applications, I think a discussion on the broader impact of the work is interesting to add the paper.

For example, the authors can discuss whether and propose a way how the error detection can help reduce safety and trustworthiness issues of the models of interest.

**Claims And Evidence:**

Yes

**Requested Changes:**

Building on the weaknesses in the previous section, here are some suggestions:

* Review that writing of some paragraphs: a more clear statement of different contribution (e.g. how the authors extend explanation methods) can only improve the clarity and impact of the work.

* It would be great to see some analysis using visual transformers.

* It would also be interesting to see results building on FullGrad or similar approaches. Alternatively, the authors should explain more explicitly why they have been discarded from the study.

* The authors can also consider a more robust evaluation (this would be a nice to have addition, but I don't see it as a requirement for acceptance).

* The paper can be slightly restructured. For example, I would suggest to move the error detection paragraph to the main text, and provide more details on it. Providing a detailed proposal and reflection on this important topic can increase the impact of the paper and the interest of the audience. In the counterparts, there are multiple redundant figures in the different sections of the main paper. For example, for the quantitative evaluation results, I would suggest for example to keep figure 11, and move figure 17 from the appendix, and move the remaining curves to the appendix. Finally, I think it would be interesting to provide more details on the user study in the main text.

**Strengths And Weaknesses:**

Strengths:
* The topic of focus in the paper is important and of interest to the community.
* The paper is generally well structured and clear.
* The analysis conducted by the authors is thorough, studying the different combinations of models and approaches, and properly isolating the different factors that can impact the quality of the explanation.
* The discussion and proposal on evaluation from the perspective of detection models is of interest, and identifies important factors to track in this setting.
* The user study is well designed, and conducted with a particular consideration for different sources of biases.
* The toolkit provided with the paper is well structured and documented, with separate components for explanation, evaluation, visualization and error analysis. It seems easy to reuse, but I haven't tested it myself.

Weakness:
* An important family of architectures seems missing from the study. Given the recent and growing interest in transformers, and the success of visual transformers for different vision tasks, including detection (see for example Carion, Nicolas, et al. "End-to-end object detection with transformers." European conference on computer vision 2020 and Beal, Josh, et al. "Toward transformer-based object detection."  (2020)), excluding them from the study seems a limitation.
* The authors also omit more recent explanation approaches, such as FullGrad (Suraj Srinivas and Francois Fleuret.  "Full-gradient representation for neural network visualization".  Advances in Neural Information Processing Systems (NeurIPS) 2019) without proper explanation or comparison. The authors state in the section 3.1 that "[guided backpropagation] is a simple and widely-used approach compared to other methods" and that "[integrated gradients] satisfy the implementation and sensitivity invariance axioms that are failed by various other state-of-the-art interpretation methods", but haven't given more details on these counterparts.
* It has been observed in previous works that score-based pixel removal metrics could be hard to interpret, and can give scores to meaningless perturbations (e.g. random) that are of comparable scale to useful saliency maps based perturbations. Moreover, given the variability of behavior across classes, images and even different instances of the same category in a single image, aggregating metrics can be misleading. It can be more rigorous to provide statistical summaries based on per image/instance pairwise rankings of methods. Both aspects are discussed for example in Grabska-Barwinska, Agnieszka, et al. "Towards Better Visual Explanations for Deep Image Classifiers."  2021.
* The contributions listed in the introduction are over-stated. For example, the error detection is only very briefly discussed in the appendix and the extension of the explanation methods to detection is not (or not well) described. The paper could benefit from some restructuring to increase clarity and impact (see suggestions below).
* [Minor] Limited novelty, as the paper builds on existing explanation methods, but the easiness of use of the toolkit as well as the discussion of the different aspects of the work, and especially the evaluation of explanations, from the perspective of object detection make the contribution interesting.
* [Minor] Some passages need rewriting to improve the clarity of the work. Examples:
[P5] "By tracking the output box corresponding to detection explained the target neuron is selected."
[P6] "Section 4.2 provides the quantitatively evaluates different detector and explanation method combinations."
[P7 - end] "Evaluating detector explanations quantitatively provides immense understanding on selecting the explanation method and"

---

### Review · Reviewer_QCaY · 2023-02-12

**Summary Of Contributions:**

This paper proposes a way to compute 'explanations' for object detectors, i.e., models that output bounding boxes to localize an object in the scene as well as providing a prediction for the particular object that has been identified in the bounding box. The paper proposes to use saliency maps, i.e., approaches that give a relevance score for each dimension of the input towards the output prediction as primitives for performing explanations. At a high level, the saliency maps selected in this work are gradient based ones, so the map corresponds to the derivative of some output with respect to the input. The proposal here obtains this gradient for each object classification, as well as each of the corresponding points in the bounding box around that object. The paper also proposes a way to aggregate these saliency maps (five per object), into a single canonical map that explains the decision of the object detector for a given image. A series of different evaluations are then conducted to justify the performance of the explanation methods.  Overall, the paper makes a commendable effort towards explaining object detectors.

**Audience:**

Yes

**Broader Impact Concerns:**

None.

**Claims And Evidence:**

No

**Requested Changes:**

Here I provide feedback to the authors on how they can strengthen this work.

- **Exposition**: Section 3.1 is much too short and does not actually describe the dext approach. An end-to-end scheme should be discussed here. In addition, Figure 3 provides an overview of the flow, but is unclear to me how the last 3 steps in that figure are actually achieved. The authors should provide, in detail, either in the appendix, or the main draft (ideally this second option), how exactly these saliency maps are computed.

- **Saliency Methods and Faithfulness**: The reliability of saliency approaches is still a hotly debated question and under strict questioning. As it stands it still unclear whether these approaches are actually effective. As a matter of fact, the evidence against the approaches used in this paper continues to accrue. For example see the papers: "Do feature attribution methods correctly attribute features", "Do input gradients highlight discriminative features", and "Rethinking the Role of Gradient Based Attribution Methods for Model Interpretability." All these papers show that the saliency maps, including the ones used in this work, *do not* highlight the features that the model is relying on for its output. However, these approaches are often used for explaining classifiers and not object detectors, but there is no reason why these results should not hold here since the backbone models used have been shown to suffer from such issues.

Here is what I would propose to rectify the above problems:

1) I am willing to accept the paper as is if the authors would acknowledge, prominently in the text, the related work showing limitations of the methods that they have used in the work. Both for the evaluation approaches, and for the methods themselves.
2) A series of alternative experiments that demonstrate the effectiveness of the saliency maps selected in this new setting. Asking for new experiments is time/cost prohibitive, so I understand that this is an extra burden on the authors. However, there is too much evidence in the literature now pointing to the limitations of these approaches to just accept this paper as is.

**New Experimental Evidence**
I think any of the following experiments would provide evidence to backup the claims in this paper:
1) A toy setting where the ground-truth saliency map is known for each object in the scene. The authors can the train models rely on a synthetic signal in the dataset that the detector should rely on for 100 percent performance. The authors can then compare the saliency maps computed to the ground  truth saliency maps since they have forced the model to rely on a synthetic signal. See the papers I cited in the previous section for additional details. The paper, "Sanity Simulations for Saliency Methods" is a good place to look for how to construct this kind of toy experiment. The critical thing here is that it must be a task where the ground truth saliency map is known ahead of time, **and**, one must be able to train a model to rely on this ground truth signal.
2) Same experiment as above but for adversarially robust detectors to see whether the issue raised in the paper "Do input gradients highlight discriminative features", translates to the object detection setting.

I would be happy to clarify any of the points I raised in this section.

**Strengths And Weaknesses:**

First, I'd like to apologize to the authors for a delayed review, especially in light of the critical nature of my review, and the changes I'd be asking for.

## Strengthens

- As far as I can tell, this is one of the first works to address the task of explaining object detectors in a comprehensive way.
- The authors make a good faith effort towards assessing the scheme they propose in a variety of ways. First, they conduct a flipping experiment, and second perform a user study to assess whether users are satisfied with the output of an explanation method.
- The open source toolkit is quite helpful, and should be useful for others in the community.

## Weaknesses
While I commend the authors for their contributions, I think their are pretty severe limitations of the work as it current stands.
- **Exposition**: It was difficult for me to understand how the saliency maps are computed especially for $x_\mathrm{min}$, $x_\mathrm{max}$, $y_\mathrm{min}$, and $_\mathrm{max}$. The last paragraph of Section 3.1 is the only one providing a high level overview of how this is done. That is currently unclear. Specifically, the authors should clarify in this section what the exact scalar output is that is being used to compute the saliency maps for those each coordinates. In addition, there is no discussion of how the canonical representation is commuted in the main text. Overall, Section 3.1 is much too sparse to provide a useful overview of the dext method proposed in this work.

- **The flipping explanation**: This paper does a commendable job of evaluation of the saliency map approaches that it uses. However, I don't believe any of the approaches used to evaluate these methods get at any meaningful measure of explanation quality. First, there are now well known critics of the Samek et. al. flipping experiments. See the following papers: 1) Sanity checks for saliency maps, 2) Sanity checks for saliency metrics, 3) Sanity Simulations for Saliency Maps, and 4) Evaluating feature importance estimates. The crux of the discussion here is that a saliency method that is not faithful to the underling detector being explained will still be able to get a high score on the flipping experiment. However, these methods will not out perform say randomly zeroing out random coordinates of the input.

- **Issue with user studies on explanations**: I commend the authors here again for performing a user study. However, there is a critical flaw here too. First, it seems like the authors asked the users: "Which Robot's explanation is better?" However, it is now well-known that just because a user likes a particular explanation type does not mean that the model is actually relying on those portions of the image to detect the object in the bounding box. The key metric of interest is whether the explanation method is communicating to the user which features the model is relying on. However, whether the user 'likes' that explanation or not is orthogonal to the faithfulness issue. Again, this metric does not tell us whether smoothgrad, GBP, and the other saliency maps are effective methods.

Overall, I think this is commendable work. However, the issues above give me high reservation about he usefulness of the dext approach and insights in this paper.

---

### Review · Reviewer_JPDP · 2023-02-26

**Summary Of Contributions:**

This paper focuses on explaining salient part of images for object detection. Authors proposed a framework which supports multiple types of object detectors and supports explaining both classification and regression part of detectors. Authors extended existing gradient-based methods to explain and visualize multiple detection outputs. On top of the proposed method, authors designed and carried out a series of quantitative and qualitative methods to evaluate the explainability of detectors.

**Audience:**

Yes

**Broader Impact Concerns:**

No concerns.

**Claims And Evidence:**

No

**Requested Changes:**

Please see the weakness.

**Strengths And Weaknesses:**

Strengths:
* The proposed analysis covers multiple different detectors, and in great detail including both classification and regression (even include 4 edges of the bounding boxes).

Weaknesses:
* Some of the details in this paper is not very clear. For example, when discussing the "Realistic Evaluation Setting", authors mentioned "matched to the interest detection by checking the same class and an IoU threshold greater than 0.9". This is not very clear to me what are the "interest detection".
* The discussed detectors are mostly old ones (especially Faster RCNN and SSD which are almost 6 years old). Some of the recent detectors are not included such as anchor-free detectors and transformer-based detectors.
* It's not super clear to me how the proposed method can identify the reason of detector failure. The Figure 26 seems to be a cherrypick to me, and it's not clear to me that why a dog with a long tail would be a cat.
* In Figure 8.c: why the AAUC differs a lot on y_min vs y_max?
* The section 4 is a bit messy and includes too many details. I feel it somewhat misses a "core observation". Also it's not very clear to me whether it is helpful to provide so many different settings (e.g. single-box vs realistic settings).
* Page 7, the section 4.2 seems to be incomplete, as the sentence ends with "and".

---

### Decision · Action_Editors · 2023-04-04

**Recommendation:** Reject

**Comment:**

Providing a library for providing saliency maps for object detection methods is a nice contribution for the community.

However, for the reasons stated above and in the reviews, the reviewers and I feel that there is insufficient evidence in the paper to support the claim that such methods can be used to reliably diagnose detection failures.

Note that TMLR does not have a "major revision" recommendation option. If the authors believe that they can address the reviewer's concerns with additional evidence then I would be willing to receive a revised version of the paper. If this were done, I would strongly suggest shortening the paper to 12 pages, and increasing the clarity of the explanation of the methods and main conclusions. Note, the revision would go through another full round of reviews.

**Audience:**

There would likely be members of the TMLR audience interested in trying out the DExT toolbox for their own detectors.

There are a couple of areas where the paper could be further improved to increase potential interest further:

(1) All reviewers noted lack of clarity in the manuscript. While the revised version is improved, I still feel it could be improved further. Section 3 omits details on how the explanations and multi-object visualizations are computed. Section 4 is quite long, and it is not easy to find the main conclusions. I think compressing this section, highlighting the key findings, and moving some redundant/supporting figures/text to the Supplementary would help.

(2) The paper studies some, now slightly older, detector designs, and not, for example, Transformer-based approaches which are popular. This is fully acknowledged in the paper, however including these would increase interest.

**Claims And Evidence:**

The paper presents an toolkit (DExT) that provides gradient-based saliency methods to provide explanations of the predictions of object detectors. The paper adapts these methods from the classification to the detection setting, and performs a study of the saliency maps of different detectors provided by different saliency methods. Finally, the paper proposes different ways to aggregate saliency maps from multiple objects into a single canonical representation for the image, and performs a user study to see which is best.

The contribution of the toolkit itself, the claim of a simple extension of gradient-based methods to object detection, and multi-object visualization are substantiated.

However, the reviewers all raised concerns about the fidelity of explanations from saliency based methods. The reviewers point to multiple studies that highlight the reliability of saliency methods, and studies that demonstrate that score-based pixel removal metrics (used to evaluate the saliency methods here) have been shown to be unreliable. One reviewer suggests to perform a toy experiment to demonstrate that the methods presented recover correctly a ground-truth saliency map. Currently it seems that there is insufficient evidence to conclude that "our work reveals some major conclusions about object detector explainability". Similarly multiple reviewers are not convinced that there is sufficient evidence that the methods presented can "identify reasons for the detector failure" given challenges with interpreting saliency methods. While the user study is a commendable addition (and potentially useful for designing features for the toolbox), assessing whether a user likes an explanation does not address whether the explanation is of high fidelity.